# xVal: A Continuous Number Encoding for Large Language Models

## Abstract

Large Language Models (LLMs) have not yet been broadly adapted for the analysis of scientific datasets due in part to the unique difficulties of tokenizing numbers. We propose xVal, a numerical encoding scheme that represents any real number using just a single token. xVal represents a given real number by scaling a dedicated embedding vector by the number value. Combined with a modified number-inference approach, this strategy renders the model end-to-end continuous when considered as a map from the numbers of the input string to those of the output string. This leads to an inductive bias that is generally more suitable for applications in scientific domains. We empirically evaluate our proposal on a number of synthetic and real-world datasets. Compared with existing number encoding schemes, we find that xVal is more token-efficient and demonstrates improved generalization.

## 1 Introduction

Even as Large Language Models (LLMs) exhibit sophisticated behavior in the generation and analysis of textual data, the scientific community has seen little success in applying these models to datasets consisting mostly of numerical values. LLMs have historically struggled to solve simple arithmetic problems such as multi-digit multiplication (Dziri et al., 2023) and have a tendency to "confabulate" answers (OpenAI, 2023; Frieder et al., 2023). Standard LLM tokenization schemes do not inherently capture the precise quantitative properties that distinguish numerical data from other natural language inputs (Testolin, 2023; Choi, 2021). Recent work exploring Chain-of-Thought reasoning in LLMs has shown improved performance on commonsense reasoning tasks such as arithmetic or mathematical word problems (Nye et al., 2021; Wei et al., 2023; Liu & Low, 2023; Imani et al., 2023), but such methods have limited applicability in making predictions about scientific datasets without highly domain-specific context.

Recent work has explored several potential improvements for encoding numerical information as inputs to language models (see Thawani et al. (2021) for a review). For instance, numbers can be encoded digit-by-digit, in scientific notation format, or in base-10 format. (Jiang et al., 2020) maps numbers onto a finite set of "prototype numerals", while (Sundararaman et al., 2020) enforces constraints such that the cosine distances between the embeddings of numbers reflects their actual mathematical distance. Transformers that use such encodings have been shown to successfully solve various mathematical problems, such as linear algebra problems including matrix multiplication (Charton, 2022).

Despite these improvements, many challenges remain unresolved. Language models are known to exploit shortcuts and spurious correlations in the data (Tu et al., 2020; Liu et al., 2022; Dziri et al., 2023) and still struggle with interpolation and out-of-distribution generalization in mathematical problems and in scientific domains (Grosse et al., 2023; Anil et al., 2022). Functions appearing in such domains are often continuous or smooth, with certain exceptions such as points of criticality. Similarly, transformer architectures applied to vision and audio domains (e.g., Dosovitskiy et al., 2020; Garg et al., 2022) typically treat numbers continuously without tokenization (see however Copet et al., 2023; Chen et al., 2020b), but these models typically require highly structured inputs, and cannot be applied to datasets with arbitrary sequences of text and numbers. On the other hand, when encoding numbers as text, LLMs are inherently discontinuous in both the encoding and decoding stages. While discrete models can (and do) learn to approximate continuous func-

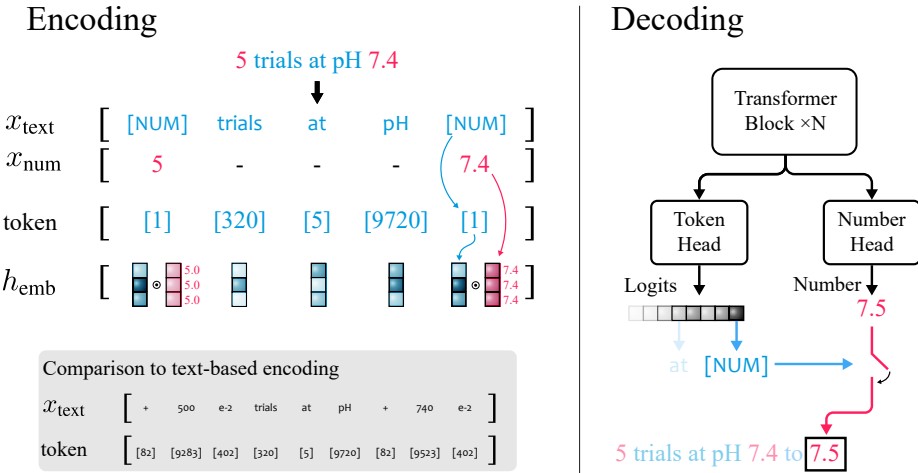

Figure 1: A simplified example illustrating the XVAL number encoding and the modified number inference paradigm. On the left, XVAL is contrasted with the P1000 text-based numerical encoding scheme. On the right, we illustrate how numbers are addressed within the decoder.

tions (d'Ascoli et al., 2022), this can be more challenging and less sample efficient compared to models that have continuity built-in by construction, as in many non-parametric regression models (Wasserman, 2006). In order to overcome this inherent challenge, it is necessary to impose the appropriate inductive bias based on our knowledge of the continuous nature of numbers.

We introduce XVAL, an inherently continuous method of encoding numerical values in Large Language Models. By encoding the magnitude of numerical values multiplicatively and orienting them in a learnable direction within the embedding space, XVAL substantially changes how numbers are processed and interpreted by transformer architectures. This leads to an encoding scheme with a single vocabulary element that also encodes every number as a single token. XVAL is therefore both token-efficient and has minimal vocabulary footprint.

Coupled with a modified number-inference paradigm, XVAL allows a transformer model to be continuous (or smooth given smooth non-linearities) when considered as a map between the numbers of the input string and those of the output. We expect that this leads to a better inductive bias when the functions being approximated are continuous or smooth. We evaluate XVAL on a number of synthetic and real-world scientific datasets and compare with existing number encoding schemes. We demonstrate that XVAL is both more token-efficient and exhibits better interpolation properties.

### OUR CONTRIBUTIONS

- We introduce XVAL, a novel approach for encoding numerical values in Large Language models. Compared to existing number encoding schemes, XVAL is both token-efficient (every number is encoded as a single token) and has a minimal vocabulary footprint (a single number token).
- We introduce a modified number inference scheme that, when used in conjunction with XVAL, renders transformer models continuous as a function of the numerical values appearing in the text.
- We evaluate XVAL and a number of existing number encoding schemes on several synthetic and real world datasets. We demonstrate that XVAL consistently provides better interpolation properties and is more compute-efficient than prior work.

## 2 METHODS

In this section, we describe the details of the XVAL number encoding as well as the number inference paradigm of our model.

## 2.1 XVAL: A CONTINUOUS NUMBER ENCODING

Instead of using different tokens for different digits or composite numbers, XVAL embeds numerical values directly along a specific learnable direction of the embedding space. A diagram of this procedure can be seen in Fig. 1. Specifically, given a string input $x$ comprising both numbers and text, we first parse $x$ to extract all the numerical values and collect them in a separate list $x_{\text{num}}$. We then construct a new string $x_{\text{text}}$ by replacing all numbers in $x$ with a designated token [NUM] that acts as a placeholder for numerical values. We tokenize and embed $x_{\text{text}}$, arriving at $h_{\text{text}}$. We then multiply the embedding of each appearance of the [NUM] token with its associated numerical value in $x_{\text{num}}$. This process can be done efficiently by defining a new list $h_{\text{num}}$ by scattering $x_{\text{num}}$ to have the same length as the tokenized $x_{\text{text}}$ and inserting a 1 for any token other than [NUM]. The final embedding of the sample is $h_{\text{emb}} = h_{\text{num}} \times h_{\text{text}}$, which is then fed to the transformer trunk.

This encoding process can be performed both for masked language modeling (MLM) and autoregressive (AR) generation. During training, in cases where MLM is used, we simultaneously mask both $h_{\text{text}}$ and $h_{\text{num}}$, i.e., if the token being masked is a [NUM] token, we replace the corresponding number in $h_{\text{num}}$ with 1.

Continuous embeddings have been previously proposed for use in attention mechanism in the context of speech recognition Chorowski et al. (2014).

**Implicit normalization via layer-norm.** In our implementation, the multiplicative embedding of XVAL is followed by the addition of a positional encoding vector and then a layer-norm in the first transformer block. The effect of the layer-norm is to normalize the embedding of each token on a per-sample basis. In our experiments, we use additive positional encodings and therefore the result of the layer-norm is to normalize the sum of the vector associated with the [NUM] token and the positional encoding vector. When the positional embeddings are not collinear to the embedding of the [NUM] token, layer-norm scales the vector associated with the [NUM] token such that its magnitude is effectively passed through a non-linear rescaling function. Indeed, denoting $u \in \mathbb{R}^d$ as the embedding of [NUM], $p \in \mathbb{R}^d$ as the positional embedding, and $x \in \mathbb{R}$ as the scalar to be encoded, and assuming for simplicity $u \cdot p = 0$ with $\|u\| = \|p\| = 1$, we have

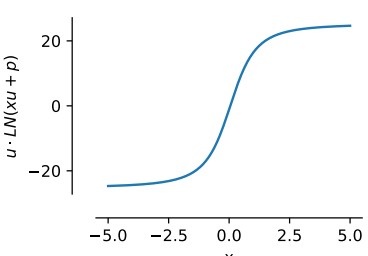

Figure 2: Value of the embedding of the number $x$ after layer-norm, projected onto the direction of the [NUM] embedding vector.

$$u \cdot \frac{xu + p}{\|xu + p\|} = \frac{x}{\sqrt{1 + x^2}},$$

such that the value $x$ is still encoded in the same direction $u$. Figure 2 shows that such a property approximately holds empirically up to a constant after training, and we found these curves to be near-identical for any positional embedding.

This normalization property implies that the dynamic range of XVAL is more limited than those of other text-based encoding schemes. In the experiments of this paper, we normalize numbers in the text corpus such that they fall within the range $[-5, 5]$ as a preprocessing step before training.

## 2.2 NUMERICAL VALUE INFERENCE

XVAL defines an embedding that is continuous in the numerical values of the input. However, if we use a multi-class classification task as our output and training algorithm, the model as a whole will not be end-to-end continuous when considering the map from the input numbers to the output numbers. For this reason, we treat numbers separately at the output layer. This process is illustrated in the right-hand portion of Fig. 1.

As is standard practice in transformer-based language models, we define a token head that outputs a probability distribution of the tokens of the vocabulary. However, since our formalism replaces numbers with the [NUM] token, this head does not carry any information about the number value. We therefore introduce a new number head with a scalar output, trained via mean squared error (MSE) loss, to recover the numerical value associated with each instance of the [NUM] token. For

any input, we first look at the output of the token head. If the generated token is the [NUM] token, we then look at the number head to fill in the value for this token. As shown in Section 3, since the transformer is now end-to-end continuous when inferring numerical values, it performs better when interpolating to previously unseen values.

## 3 EXPERIMENTS

In this section, we evaluate the performance of XVAL and highlight its strengths and weaknesses compared to existing numerical encoding algorithms. In particular, we look at three datasets: a synthetic dataset of arithmetic operations, a dataset of global temperature data, and a dataset of planetary orbit simulations.

For our transformer models, we use an architecture based on GPT-2 (Radford et al., 2019). Details of our specific architecture are included in Appendix A). We explore the effects of various architectural design choices in Appendix B.4.

Table 1: Comparison of XVAL with four other number encodings. XVAL is more token-efficient and has a minimal vocabulary footprint. Vocabulary size differs from Charton (2022) because we only consider exponents from 1E-8 to 1E+8.

| Encoding | Tokens | $-6.02 \times 10^1$ | Tokens per number | Vocabulary Size |
|---|---|---|---|---|
| P10 | {±, d, E±d} | [-, 6, 0, 2, E-1] | 5 | 28 |
| P1000 | {±, ddd, E±d} | [-, 602, E-1] | 3 | 918 |
| B1999 | {±ddd, E±d} | [-602, E-1] | 2 | 1816 |
| FP15 | {±ddd E±d} | [-602 E-1] | 1 | 28800 |
| XVAL | {[NUM]} | [NUM] | 1 | 1 |

**Comparison with other number encodings.** We compare the performance of XVAL with four other number encodings, following the notation of Charton (2022). In these encodings, numbers are first processed into the format ±ddd E±d. The encodings are then determined by which parts of this format are encoded as single or multiple tokens. These range from encodings with limited vocabulary size but high number of tokens per number, leading to longer encoded sequence lengths (e.g., P10), to those with very large vocabulary footprints but only one token per number, leading to shorter encoded sequence lengths (e.g., FP15). XVAL provides a minimal vocabulary footprint and uses just a single token per number, leading to the shortest sequence lengths. A summary of these encodings and an example can be seen in Table 1.

Number encodings that do not lead to a fixed number of tokens for all numbers (e.g., learned Byte Pair Encoding (Gage, 1994) used in GPT-2 (Radford et al., 2019)) can lead to erratic behaviors where the transformer learns spurious correlations that exist between the length of the encoded numbers in the dataset. An example of this type of behavior is shown in Appendix B.3.

### 3.1 LEARNING ARITHMETIC

Simple arithmetic problems have acted as a test bed for probing the mathematical reasoning abilities of language models (Dziri et al., 2023). In this section, we investigate the effect of the number encoding scheme on the ability of language models to perform multi-digit multiplications as well as multi-operand mathematical operations. Multi-digit multiplication is a notably challenging task for even the largest LLMs (Borji, 2023). Dziri et al. (2023) show that GPT-4 achieves only 59% zero-shot accuracy on three-digit multiplication problems, while its accuracy for four- and five-digit multiplication drops to 4% and 0%, respectively.

Table 2 reports the $R^2$ scores for multi-digit multiplication problems on several language models designed to handle numerical values. All number encodings generally perform well on this task. However, we find that some encoding schemes (P10 and FP15) show a tendency to yield a small percentage of highly erroneous predictions in some contexts, thereby reducing the $R^2$ score, while XVAL does not produce such outliers.

For a more challenging arithmetic task, we designed a dataset of multi-operand mathematical operations. We used random binary trees combining a fixed number of operands (2, 3, or 4) using the binary operators of addition, subtraction, and multiplication. build a dataset in which each sample is an arithmetic statement such as `((1.32 * 32.1) + (1.42-8.20)) = 35.592`. We then processed the samples according to the processing requirements of each number-encoding scheme. The task is evaluation of the expression on the left-hand side of the equation, implemented as a mask completion, where the right-hand-side number is masked. Table 3 shows the adjusted $R^2$ scores results on this task. XVAL performs remarkably well on this task.

Table 2: Adjusted $R^2$ scores calculated between predictions and true values for the different encodings on various arithmetic datasets. (Higher is better; $R^2 = 1$ is the theoretical maximum.)

| Encoding | 3-digit Multiplication | 4-digit Multiplication | 5-digit Multiplication |
|---|---|---|---|
| P10 | 0.9989 | 0.6071 | 0.9439 |
| P1000 | 0.9997 | 0.9783 | 0.9991 |
| B1999 | 0.9998 | 0.9984 | 0.9997 |
| FP15 | 0.7119 | 0.9959 | 0.9980 |
| XVAL | 0.9986 | 0.9975 | 0.9958 |

Table 3: Arithmetic evaluation task of random binary trees combining different numbers of operands with addition, subtraction, and multiplication. $R^2$ measured between true expression value and transformer prediction.

| Encoding | 2 operands | 3 operands | 4 operands |
|---|---|---|---|
| P10 | 0.998 | 0.996 | 0.992 |
| P1000 | 0.991 | 0.990 | 0.991 |
| FP15 | 0.993 | 0.981 | 0.935 |
| XVAL | 0.99998 | 0.99994 | 0.99998 |

Arithmetic experiments alone are not sufficient for fully evaluating the mathematical abilities of language models. The samples in these datasets are often short sequences and the underlying data manifold is low-dimensional. These problems therefore do not push the boundary of what is computationally possible with LLMs.

In the remainder of this section, we consider experiments in more complex settings and much longer sequences. The goal of the next two subsections is not to construct state-of-the-art models in their respective domains, but rather to compare the performance of language models with different number encoding schemes in more complicated real-world scenarios.

## 3.2 TEMPERATURE FORECASTING

As an example of real-world scientific analysis, we look at the task of temperature forecasting. In this experiment, we construct a dataset as a subset of the ERA5 global climate dataset (Hersbach et al., 2020). For simplicity, we only focus on the surface temperature data (T2m field in ERA5). We split the dataset into individual samples, where each sample includes 2–4 days of surface temperature data (normalized to have unit variance) as well as the latitude and longitude from 60–90 randomly selected reporting stations. We also include the time of the first included timestep. We encode the coordinates by using the sine of the latitude and the sine and cosine of the longitude such that we preserve the periodicity. Similarly, we encode the time of year and time of day using the sine and cosine of the position along the 24 hour and 365 day cycles. We include all this information in a JSON format as follows[1]:

```
{'description':{'coords':[[1,-.32,.95] ... [.96,.61,.79]],
'start':[0,1,-.026,-1]}, 'data':[[-2.6,-2.6 ... -3.2,-3.1,-3]]}
```

---

[1]For demonstration purposes, we show a few digits per number, but for both scientific datasets, all numbers are floating point numbers. For the text-based encodings, this text string is then processed according to the procedure described above.

The `coords`, `start`, and `data` correspond to the reporting station coordinates, the time of the first sample, and the normalized temperature data, each reported separately per station and then concatenated in the data list. In this way, the model needs to parse both the textual aspects of the sample (e.g., where the commas appear to separate different parts of the data) as well as the numerical values. Furthermore, as is often the case with JSON-formatted data, the data does not have a causal format. We therefore train the language models using an MLM approach instead of the more common AR approach. We evaluate the performance of the different numerical encodings

Table 4: Performance (in MSE) and runtime of the different encodings on predicting the temperature for the next time step. "Equal Samples" columns refer to all models being trained for 500k iterations. Training was performed on 4 Nvidia H100 GPUs using Pytorch Distributed Data Parallelism.

| Method | Equal Samples | | Equal Tokens | | Equal Runtime | |
| --- | --- | --- | --- | --- | --- | --- |
| | Loss | Runtime | Loss | Runtime | Loss | Runtime |
| P10 | 73 | 2d 22h | 73 | 2d 22h | 73 | 2d 22h |
| P1000 | 20 | 2d 2h | 23 | 3d 10h | 21 | 2d 22h |
| B1999 | 20 | 20h | 19 | 2d 23h | 19 | 2d 22h |
| FP15 | 2.14 | 19h | 1.76 | 3d 12h | 1.85 | 2d 22h |
| xVAL | **1.75** | **9h** | **1.62** | **1d 15h** | **1.51** | 2d 22h |

on the task of predicting the next temperature timestep for all reporting stations simultaneously in a held out test set. We do so by masking the tokens (and numbers, if applicable) of all the data associated with the final timestep. Because the temperature data is provided separately per station, the masks are scattered throughout the input data and are not all simply at the end of the sample.

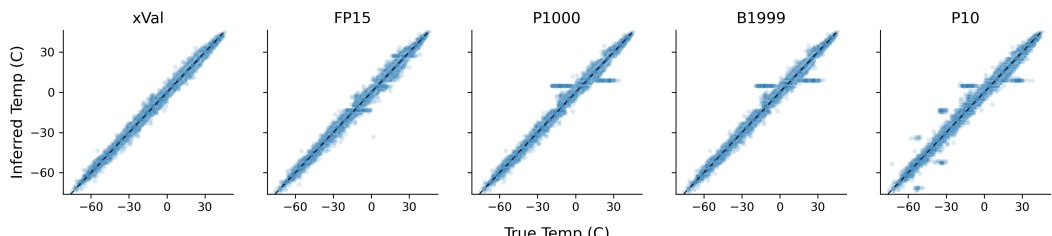

Figure 3: Performance of the encoding schemes predicting the temperature of the next timestep.

Table 4 shows the results of this experiment. XVAL provides the best performance while taking considerably less compute time.

This task exemplifies one of the shortcomings of text-based encoding schemes: they can take advantage of spurious correlations in the data. In this case, P10, P1000 and B1999 have a tendency to predict normalized temperature $\pm 0.1$, which manifest as extended protrusions in Fig. 3. This is due to the over-abundance of this number in the dataset compared to other numbers, as seen in Fig 4. While individually, `100` and `E-3` are the most common numbers and exponents in the dataset, when combined, `100E-2` is much more frequent than `100E-3`. This explains why FP15, which encodes the digits and exponents as one token, does not get confused in this case. It also implies that the model has failed to learn the correct joint distribution of the numbers. In these cases, because of the tokenization scheme, the length of the tokenized samples are very long, averaging around 8000 and 5000 tokens respectively for P1000 and P10 (compared to 1800 tokens for FP15 and XVAL). The poor performance in these models might therefore be due to the the challenges of modelling long-range interactions (Qin et al., 2023).

For more details on the performance of the different encodings, as well as comparison with some non-transformer baselines, see Appendix B.1. In Appendix B.3 we look at the performance of a BPE tokenizer on this task and demonstrate how LLMs can exploit the tokenized length of the number. In Appendix B.1.3 we train fine-tune these models on a simple binary classification task and compare their performance.

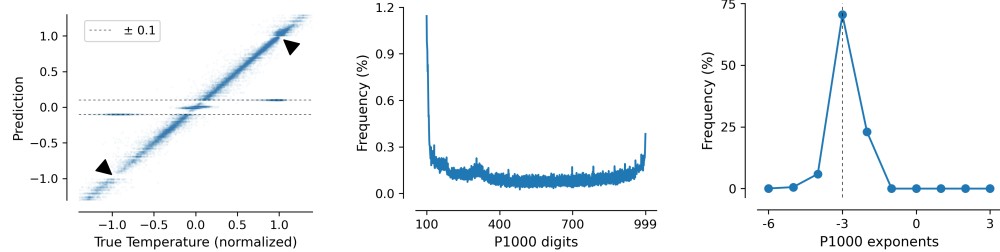

Figure 4: A failure mode of text based encoding scheme (left). Because of the distribution of the numbers in the training set (center and right), numbers that are close to $\pm 1$ (denoted by the black arrows) get misclassified as `100E-3`, i.e. 0.1, the combination of the most common digit and the most common exponent in the dataset.

### 3.3 PREDICTING PLANETARY ORBITS

We then compare the performance of the various number encoding schemes on a simulated dataset of planetary orbits. We construct a dataset consisting of planetary motion simulations generated by the REBOUND N-body codebase (Rein, H. & Liu, S.-F., 2012) and integrated using IAS15 (Rein & Spiegel, 2015). The dataset consists of 1.25 million samples, split into 80%, 10%, 10% for training, validation, and test. Each sample consists of simulation parameters (mass and orbit properties of each planet and the simulation timestep size) as well as a sequence of $(x, y)$ positions for each planet, organized in a JSON format. The details of the simulation are provided in Appendix B.2. A typical sample in this dataset is given by:

```
{'description':{'planet0':{'m':2.38, 'a':2.96, 'e':1.73},
'planet1':{'m':1.35, 'a':2.96, 'e':1.73}, ... , 'stepsize':0.2},
'data':[[[2.60,-0.75],[0.81, 0.42]],[[2.63,-0.63],[0.70,0.60]]...]}
```

We pretrain the models using MLM and evaluate the models on the task of inferring the simulation parameters, specifically the simulation timestep $\Delta t$, and the semi-major axis, eccentricity and mass of the first planet $(a_1, e_1, m_1)$ by masking the appropriate locations. The quantities $\Delta t$ and $a_1$ in the training corpus take values that are either discrete or are sampled from intervals with gaps. This property makes these quantities a good testing ground for interpolation generalization.

Table 5: Performance of the different encodings on the planetary motion inference problem. Here, OoD implies evaluation on samples where the quantity was not seen in the training corpus. The percentages in brackets denote the fraction of the predictions that could not be parsed as numbers. When not specified, this fraction was less than 0.01%. (†) The poor performance here is because of a number of outliers that are being mis-classified.

| Method | $a_1$ | $a_1$ (OoD) | $e_1$ | $\Delta t$ | $\Delta t$ (OoD) | $m_1$ |
|--------|-------|-------------|-------|------------|------------------|-------|
| P10    | $7.6 \times 10^{-4}$ | 0.0076 (1%) | 0.20 | **0.0** | 0.0036 | 1.5 |
| P1000  | $4.5 \times 10^{-6}$ | 0.048 | 0.0067 | **0.0** | 0.011 | 0.74 |
| B1999  | $\mathbf{3.6 \times 10^{-6}}$ | 0.11 | 0.0057 | **0.0** | 0.022 | 0.44 |
| FP15   | $4.0 \times 10^{-6}$ | 0.050 | $\mathbf{3.6 \times 10^{-4}}$ | $0.0065^{\dagger}$ | 0.0075 (0.2%) | **0.37** |
| XVAL   | $6.4 \times 10^{-5}$ | **0.0010** | 0.0020 | $6.6 \times 10^{-5}$ | **0.0021** | 1.4 |

The results of this test are presented in Table 5. In the numerical encoding schemes other than XVAL, we see an overall inverse relationship between performance in- and out-of-distribution. For example, P10—the encoding with the fewest vocabulary elements—provides the worst in-distribution performance but is best on out of distribution tasks. This is an example of the bias/variance trade-off applied to the number of vocabulary elements.

In comparison, we see that XVAL provides the best out-of-distribution performance while staying competitive in-distribution (with one exception). The out-of-distribution performance of these encoding methods can be seen in Fig. 5. Here we see that the text-based encodings, with the exception

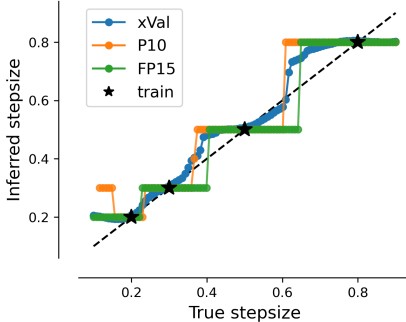 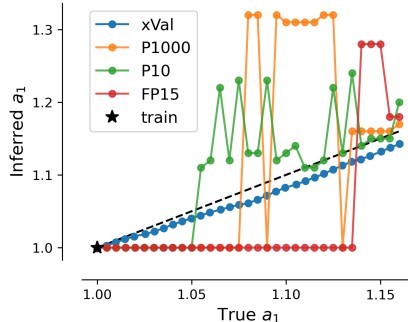

Figure 5: Out of distribution generalization properties of the different number encoding schemes. Left: Inferring $\Delta t$, which takes discrete values in the training set. Right: Inferring $a_1$ which is either 1 or $> 1.16$ in the training set. Because of the generation procedure, taking $a_1 \rightarrow 1.16$ here does not result in an in-train-distribution sample.

of P10, simply do not predict any number that they did not explicitly see for this parameter in the training corpus. As expected from a function that is continuous by construction, XVAL continuously interpolates between the values seen in the training set and offers much better performance.

Figure 5 shows that the predictions coming from the text-based encodings can be discontinuous when evaluated out-of-distribution. This discontinuity has two potential sources: the discontinuous nature of the number embeddings and the argmax that is taken over the logits during inference. Since the encodings of the number tokens in text-based encodings have been shown to form continuous-looking structures (see Sec. B.5 and Power et al. (2022); d'Ascoli et al. (2022)), it is possible that the discontinuity is only a side effect of the argmax and that the logits themselves vary more smoothly. Figure 6 shows an example of the logits of the P1000 encoding when predicting the step-size out-of-distribution. Here, the color lines denote the highest-value logits, with the other logits carrying negligible weight. The dashed gray lines denote the values of the step-size seen in the training set. We see that these lines are smooth in neither small or larger scales. We expect that this is a combination of the text-based number encodings' discrete embedding schemes together with the cross-entropy training paradigm that does not incorporate number distances into the loss.

## 3.4 RESULTS SUMMARY

It is evident that embedding the magnitude of numbers directly, as in XVAL, leads to a different inductive bias than treating numbers as tokenized text. This can be clearly seen in the varying performance of these language models in different tasks. When predicting the next timestep in the temperature dataset, XVAL provides by far the best results. On the other hand, in the mass prediction on the planetary task, it fails to learn the correct relationship, along with vocabulary-sparse P10.

Where XVAL excels is in out-of-distribution performance, while the text-based encoding schemes fail to interpolate properly. The best interpolation for the text-based encodings is given by the vocabulary-sparse P10, which performs poorly on the in-distribution tasks. However, it often performs poorly when evaluated on in-distribution tasks. The the extra encod-

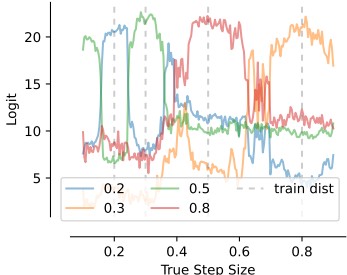

Figure 6: An example of the logits of model trained with P1000 encoding evaluated on the step-size prediction task.

ing length of P10 also makes it prohibitively expensive to deploy as can be seen in Table 4. On the other hand, FP15 provides the best in-distribution performance but it has poor interpolation properties and expensive embedding cost. Overall, XVAL provides the best mix of in-distribution and out-of-distribution performance. Moreover, it is the most computationally efficient of the encoding schemes we considered.

**Failure modes.** There are a number of ways that number inference via a large language model can fail. The language model can predict a non-numeric token in the place of the number, leading to an invalid prediction. These are denoted in the percentages in brackets in Table 5, shown only when the percentage exceeded 0.01%. This failure mode is uncommon and becomes less frequent the more the model is trained. Another failure mode is when the model exploits spurious correlations. For example, the model can learn the distribution of the digits, as discussed in the example of temperature dataset, or the length of the encoding (see Appendix B.3).

A model can also fail to learn the correct distribution. In the planetary orbits example, learning the mass of the planet is the most challenging task – all encodings struggle with this. In this task, XVAL performs uncharacteristically poorly. We suspect that this is due to the high uncertainty in estimating the mass and that a multi-modal distribution such as the categorical distribution learned by traditional LLMs would perform better. This can be seen in Fig. 7, where the predictions of P10 and XVAL are shown. While both of these models perform poorly when considering the MSE of the prediction, the multi-modal prediction of P10 would be a better starting point for capturing an uncertain distribution. We therefore suspect that generalizing the number-head such that instead of predicting a scalar for each number, it fits a mixture of Gaussians, would improve this performance. We leave explorations in this direction for future investigation.

## 4 DISCUSSION

In this work, we introduced XVAL, a continuous number encoding that makes transformer-based models end-to-end continuous when considered as a function mapping the numerical values of the input to those of the output. We demonstrated that even though XVAL is more token-efficient and has a minimal vocabulary footprint, it excels in numerical tasks and leads to superior performance, especially when evaluated on out-of-distribution samples. Because of the fundamentally different treatment of numbers across these cases, XVAL and text-based encodings lead to different inductive biases, making the choice of the best encoding method on a given dataset highly dependent on the problem under consideration.

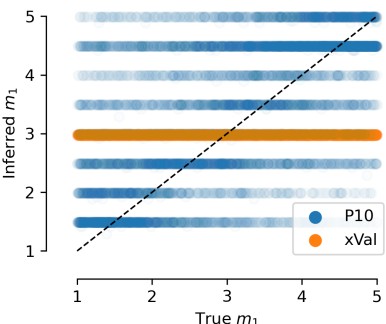

Figure 7: Different failure modes of XVAL and text-based encodings. XVAL and P10 perform equally poorly on this task, but their predictions look different.

**Future directions.** As we have seen, using the XVAL encoding scheme renders the LLM not just continuous, but also differentiable as a function of the numbers it predicts. This enables the LLM loss to incorporate not just an MSE loss, but other statistical learning schemes. For example, we can add a Gaussian Mixture Model or any other differentiable loss and train the LLM to optimize this objective. This holds the promise to improve the experiments in which XVAL underperformed in this paper.

A shortcoming of XVAL is that, because it embeds number values directly in the embedding space, its dynamic range is limited compared to text-based encodings. Very large numbers saturate the normalization, as discussed in Sec. 2, and very small numbers are negligible from the model's perspective. There are methods that allow high dynamic ranges that maintain continuity (or smoothness). One such example is to use Fourier features on the logarithm of the number. This can be considered as a continuous analog of floating point precision encoding and would drastically improve the dynamic range of the XVAL encoding.

XVAL, combined with our proposed number-inference paradigm, makes LLMs generally more suitable for applications in scientific domains. LLMs have become increasingly integrated in many scientific workflows today, enabling researchers to parse scientific language in sophisticated ways. However, their usefulness for analyzing data-heavy corpuses is currently limited. Crafting LLMs that have a better understanding of numerics has the potential to greatly increase their usefulness in scientific analysis and discovery.

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
