# A   ARCHITECTURE DETAILS

In our experiments, all the language models, regardless of encoding, adopt the main features of GPT-2 (Radford et al., 2019). That is, we use absolute position encoding and the transformer blocks have layer norms prior to the attention module and the MLP (i.e., after each residual connection). We also set the width of the MLP hidden layer equal to 4 times the width of the embedding. We deviate from GPT-2 in that we initialize all the weights of the transformer blocks with a normal distribution with standard deviation given by $(2 \times \text{fan-in} \times \text{num-layers})^{-1/2}$. The dependence of the standard deviation on the number of transformer blocks is to counteract the effect of having a series of residual connections. We also do not use any biases in the trunk of the transformer. As is standard, after the transformer blocks, we have a token-head, comprised of a single linear layer, which maps the latent embedding of each token into a distribution over the vocabulary. As in GPT-2, we tie this weight to that of the embedding matrix which maps the tokens of the input to the embedding space.

For the LLMs using XVAL encoding and an MSE number-head in addition to the token head, we promote both heads (number and token) to be MLPs with one hidden layer of width equal to the embedding dimension. This was to allow the two different prediction types (the number and the distribution over the vocabulary) to be processed separately before the final prediction. In particular, we explore the possibility of having biases for the number-head and not in the token-head in Sec. B.4.

For all of our training runs, we use a cosine learning-rate schedule with warm-up. The scheduler is adjusted such that it reaches the minimum learning rate at the end of the training run.

# B   FURTHER EXPERIMENTAL DETAILS

In our experiments, the hyperparameters for Learning Rate via a grid search coarse grid search in log space with spacing given by a factor of 2. We chose the best performance on a validation set and reported the results on an unseen test set. We have added this description to the supplementary materials section. The result of the search can be seen in tables below.

## B.1   TEMPERATURE FORECASTING

### B.1.1   EXPERIMENT DETAILS

**Dataset details.**   The ERA5 dataset (Hersbach et al., 2020) is a high-resolution, state-of-the-art global atmospheric reanalysis product provided by the European Centre for Medium-Range Weather Forecasts (ECMWF). It is the fifth generation of ECMWF atmospheric reanalyses and represents the latest advancement in the ERA (ECMWF Re-Analysis) project. The dataset covers the period from 1979 to near-real-time and is updated regularly.

In our experiment, we take only the surface temperature of the dataset (field T2m) sampled at 8 hour intervals. For each sample, we randomly choose 60–90 of the ∼1-million spatial grid points of the dataset, and include 8–16 temperature time points at 8-hour intervals (corresponding to 2–4 days), starting from a random time. We generate 1.25 million examples in this way and split it into 1 million train, 125 thousand validation and test set samples.

```
{'description':{'coords':[[1,-.32,.95] ... [.96,.61,.79]],
'start':[0,1,-.026,-1]}, 'data':[[-2.6,-2.6 ... -3.2,-3.1,-3]]}
```

The samples are individually preprocessed such that the temperature range across all samples has mean zero and standard deviation equal to 1. We also include the lattitude longitude information. To respect the periodicity of this information, we provide the sine of the lattitude and the sine and cosine of the longitude. Furthermore, we specify the starting time for each sample as the day of year and time of day. Again to respect the periodicity of these quantities, we provide the sine and cosine of the phase of these quantities.

**Architecture design hyperparameters.**   For all experiments done with this dataset, we use transformers with 6 transformer blocks, each with 6 heads and each head having width 128, resulting in a embedding width of 768 (43.5M parameters).

**Training hyperparameters.** For the equal samples training runs, we train each model for 500k iterations with batch size equal to 64 samples. For the equal tokens runs, we increase the number of iterations proportionately such that the total number of tokens seen is equal. This implies: 500k samples for P10, 820k for P1000, 1.2M for B1999, and 2.3M for FP15 and XVAL. Since there is non-numeric data in the samples, the ratio of the length of the equal tokens is slightly different from the ratio of the length of each encoding scheme's tokenization length for numbers. The other hyperparameters in this task are given in Table 6.

Table 6: Training hyperparameters for the different encodings on the Temperature Forecast dataset.

| Encoding | Learning Rate | Minimum LR | Warmup | Max Context Length |
|----------|---------------|------------|--------|--------------------|
| P10 | $2.5 \times 10^{-5}$ | $2.5 \times 10^{-6}$ | 2000 | 8222 |
| B1999 | $10^{-4}$ | $10^{-5}$ | 2000 | 1251 |
| P1000 | $10^{-4}$ | $10^{-5}$ | 2000 | 5010 |
| FP15 | $10^{-4}$ | $10^{-5}$ | 2000 | 1798 |
| XVAL | $2 \times 10^{-4}$ | $2 \times 10^{-5}$ | 2000 | 1798 |

### B.1.2 NON-TRANSFORMER BASELINES

To understand this task better, we trained a number of non-transformer baselines for comparison. These models are reported just for comparison and by no means represent the best possible non-transformer based baslines.

First, we looked at the performance of an MLP model when trained in a supervised way to predict the next time step (All stations). To deal with the varying number of locations and varying number of time-steps, we simply keep the number of locations/time-steps that is the minimum across all samples (60 locations and 8 time-steps.) We then looked at the possibility of temperature forecast based on a single reporting station (Single Station). And then on this single-station dataset, we looked at the performance on the temperature data alone (Single Station - temp all), temperature data + station coordinate (Single Station - temp + coord), and temperature data + first time step time of year (Single Station - temp + ToY).

The MLPs acting on single stations have 3 hidden layers of width 256. The MLP looking at 60 stations simultaneously is larger to validate that the poorer performance is not because of limited network size. We tried width from 256–8192 and up to 5 layers and the results remain similar.

Table 7: Temperature forecast MLP baselines

| Method | MSE Loss (C) |
|--------|--------------|
| All Stations | 2.31 |
| Single Station | 1.57 |
| Single Station - Temp only | 1.79 |
| Single Station - Temp + Coord | 1.65 |
| Single Station - Temp + ToY | 1.74 |

The results of these tests can be seen in Table 7. We see that for good performance, it is important for the model to have access to both the time of year as well as the coordinate of the reporting station. However, providing the information for multiple reporting stations at once makes the performance worse.

This implies that for the transformer model to be able to predict the temperature with MSE less than 1.7, it needs to properly parse all this information that is scattered across the different parts of the input string. XVAL was the only model to achieve MSE below that of the MLP model (Table 4) meaning that it has likely learned to leverage the temperature of other reporting stations as well.

### B.1.3 COMPARISON OF FINE-TUNING BEHAVIOR

In this section, we explore the fine-tuning behavior of the different encoding schemes in a simplified setting.[2] In this problem, we fine-tune a downstream model to predict whether or not the location of the first reporting station in the sample is located on the ocean. As this is a binary classification task, we train logistic regression on the final embedding of the transformer (the output of the last transformer block). We use 500 training samples for this task. While this problem is in principle solvable by looking at the latitude and longitude of the reporting station which is included in the data, 500 samples is not enough to learn this map. Therefore, the model needs to leverage other information in the temperature patterns to make this prediction. Table. 8 reports the performance of these models. We report the ROC AUC as a more balanced metric, since the distribution of land vs ocean on earth is not symmetric.

Table 8: Performance of the different number encoding schemes when fine-tuned on the binary task of predicting whether the first reporting station is on the ocean or on land.

| Method | ROC AUC |
|--------|---------|
| P10    | 0.580   |
| FP15   | 0.600   |
| xVal   | 0.62    |

### B.2 PLANETARY MOTION

**Dataset details.** In this dataset we use the REBOUND N-body simulation codebase (Rein, H. & Liu, S.-F., 2012) and IAS15 integrator (Rein & Spiegel, 2015) to generate a number of planetary systems (with a central mass $m_\odot \equiv 1$) and follow their orbits for a number of time points. Each planetary property is drawn from a uniform prior: the number of planets $n \in [2, 4]$, mass $m/m_\odot \in [10^{-5}, 5 \cdot 10^{-5}]$, semimajor axis equally spaced for the planets between 1 and $a_f \in [1.5, 3]$ (i.e. if 3 planets and $a_f = 1.8$ then $a_1 = 1$, $a_2 = 1.4$ and $a_3 = 1.8$), eccentricity $e \in [0, 0.1]$, and starting angle in the $(x, y)$ plane equal to zero for 30% of the samples and uniform $\theta \in [-\pi/6, \pi/6]$ for the remainder. These choices are made such that when generating the large number of samples required for training, we do not come across instabilities or collisions. Finally, we use an integration step-size sampled uniformly from $\{0.2, 0.3, 0.5, 0.8\}$.

We generate 1.25 million examples in this way and split it into 1 million train, 125 thousand validation and test set samples. We normalize the masses such that they take value between 1 and 5 and the eccentricities such that they are between 0 and 2. We then construct a JSON format sample including all of this information. A generic sample is given in this example.

```
{'description':{'planet0':{'m':2.38, 'a':2.96, 'e':1.73},
'planet1':{'m':1.35, 'a':2.96, 'e':1.73}, ... , 'stepsize':0.2},
'data':[[[2.60,-0.75],[0.81, 0.42]],[[2.63,-0.63],[0.70,0.60]]...]}
```

**Architecture design hyperparameters.** Similar to the Temperature Forecasting dataset, for all experiments, we use transformers with 6 transformer blocks, each with 6 heads and each head having width 128, resulting in a embedding width of 768 (43.5M parameters).

**Training hyperparameters.** We train each model for 500k iterations with batch size equal to 64 samples. The hyperparameters in this task are given in Table 9.

### B.3 ERRATIC BEHAVIOR OF NUMBER ENCODINGS OF UNFIXED LENGTH

In many JSON formatted datasets, the data does not follow a causal pattern, i.e. earlier entries might depend logically on latter entries. This is also the case for our JSON formatted samples. Because of this we used Masked Language Modeling (MLM) for pretraining our models. In the context of

---

[2]We are grateful to reviewer qrLU for suggesting to look beyond evaluation on data present directly in training samples.

Table 9: Training hyperparameters for the different encodings on the Planetary Motion dataset.

| Encoding | Learning Rate | Minimum LR | Warmup | Max Context Length |
|---|---|---|---|---|
| P10 | $10^{-4}$ | $10^{-5}$ | 2000 | 2707 |
| B1999 | $10^{-4}$ | $10^{-5}$ | 2000 | 1251 |
| P1000 | $10^{-4}$ | $10^{-5}$ | 2000 | 1736 |
| FP15 | $10^{-4}$ | $10^{-5}$ | 2000 | 767 |
| XVAL | $2.5 \times 10^{-5}$ | $2.5 \times 10^{-6}$ | 2000 | 767 |

MLM, number encodings that lead to encoding lengths that vary based on the number can prove troublesome both during training and during testing. During train time, the length of the encoding acts as a cue to help the model figure what the number is. This is an example of spurious correlations that LLMs are known to exploit (Tu et al., 2020; Liu et al., 2022; Dziri et al., 2023). Similarly at test time, the length of the mask can bias the model toward predicting one number or another.

As a demonstration of this feature, we first preprocessed the Temperature Forecast dataset such that every number has only two significant figures and drop leading zeros for efficiency (e.g. 0.12 → .12).[3] We then used a tokenizer that included single and double digits as well as ±, the decimal point and exponents ranging from (E-8 to E+2). In this dataset, Positive and negative floats with magnitude between 0.1 and 1 (e.g. .23 and -.34) would have encoding lengths equal to 2 and 3 and Positive and negative floats with magnitude between 0.01 and 0.1 (e.g. -.034 = 3.4E-2) would have encoding lengths 4 and 5. There are exceptions however. For example in this scheme 0.030=3E-2 has encoding length 2.

The results of this experiment can be seen in Fig. 8. We see that even though the model's overall performance is not great, it can tell with very high accuracy the numbers sign, whether or not it has absolute value greater/less than 1, or greater/less than 0.1. This is due to the fact that the model is exploiting the correlation of the numbers with the length of the encoding. We verify this by highlighting in orange the cases where in the range between 0.01 and 0.1, the number has encoding length 2, that is it does not follow the general trend mentioned above. We see that the model believes that these numbers are greater than 0.1 (which as we saw generally had encoding length 2).

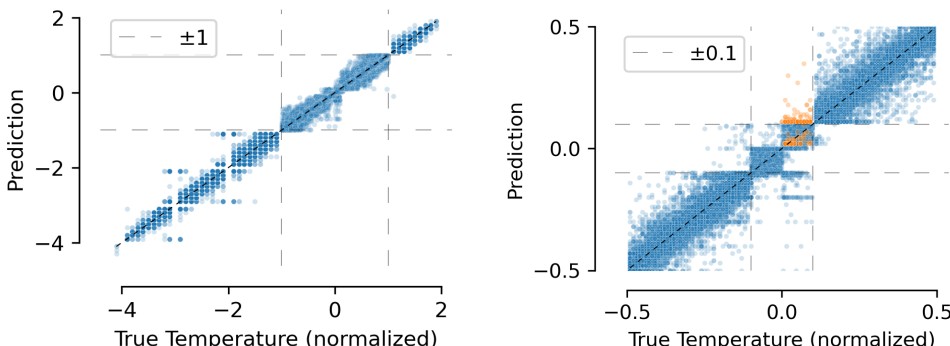

Figure 8: LLMs can exploit spurious correlations in the data. In this case, the model has learned the correlation between the number signs/values with the length of the encoding. Highlighted in orange are numbers between 0 and 0.1 that do not have the encoding length equal to 2..

## B.4 ARCHITECTURAL EXPLORATIONS

There are a number of engineering choices that we made regarding the architecture and hyperparameters of the transformer models trained with XVAL and the number head. Here, we explore the effect of these on the Temperature Forecast task. Because of the large exploration space and the high

---

[3]In the experiments of the Sec. 3, the numbers have three significant figures. Therefore the results of this section are not directly comparable to those of the main text.

amount of compute required, we do the ablation tests on a shorter run, 100k iterations compared to 500k iterations of the main text. For this exploration, we first run all of the configurations with 4 different learning rates (2.5E-5, 5E-5, 1E-4, 2E-4). We then choose the best performing learning rate for each configuration and then run each configuration two more times with this learning rate. The result of this exploration is given in Table 10.

Table 10: Ablation tests for the various design choices. Here Normal refers to min-LR/lr=0.1, Weight decay = 0.1 and MLM probability = 0.2, and the opposite dichotomy for the other choices.

| Configuration | Best Validation Loss | Learning Rate |
|---|---|---|
| Normal | $(6.8 \pm 0.2) \times 10^{-3}$ | 0.0002 |
| min-LR/LR = 0.01 | $(7.0 \pm 0.1) \times 10^{-3}$ | 0.0002 |
| First Layer Norm = False | $(6.8 \pm 0.5) \times 10^{-3}$ | 0.0002 |
| MLP Layer Norm = False | $(9.0 \pm 0.1) \times 10^{-3}$ | 0.0001 |
| MLM probability = 0.1 | $(8.2 \pm 0.6) \times 10^{-3}$ | 0.0002 |
| MLM probability = 0.3 | $(6.4 \pm 0.4) \times 10^{-3}$ | 0.0002 |
| Weight decay = 0.0001 | $(8.2 \pm 0.6) \times 10^{-3}$ | 0.0002 |
| Weight decay = 1 | $(5.3 \pm 0.3) \times 10^{-3}$ | 0.0002 |
| Trunk bias = True | $(6.2 \pm 0.4) \times 10^{-3}$ | 0.0002 |
| Num-head bias = False | $(6.9 \pm 0.1) \times 10^{-3}$ | 0.0002 |

We summarize the various configurations that we run this experiments in and their effects as follows:

- Ratio of the final learning rate of the cosine scheduler to the initial learning rate (min-LR/LR). We found decreasing this ratio from 0.1 to 0.01 does not affect performance in this experiment. But we found that it does increase stability in longer runs.

- Turning off the layer norm prior to the MLP of the first transformer block (First Layer Norm = False). This change does not affect average performance. This is not surprising since the effect of the layer norm at this stage is simply to normalize the numbers and the numbers in this dataset are in the regime where the normalization discussed in Sec. 2.1 is linear.

- Turning off the layer norm prior to the MLPs of all transformer blocks (MLP Layer Norm = False) This change had a significant negative impact on the performance of the model.

- Changing the masking probability to 10% or 30% (default is 20%). Decreasing (resp. increasing) this probability lead to performance deterioration (resp. improvement) in this experiment. However, this seems to be dependent on the dataset as in other instances 30% seems to be too high for effective learning.

- Changing the weight decay to 0.0001 or 1 (default is 0.1). Increasing this value lead to the largest improvement. However, similar to the masking probability, this seems to be dataset dependent. The effect of increased weight decay can also depend on the length of the run.

- Including a bias in the modules of the transformer block (they are absent by default). Including this bias improved performance at the cost of increased variability.

- Turning off the bias in the number head (present by default). This change did not affect the performance significantly.

## B.5 LEARNED EMBEDDINGS FOR TEXT-BASED NUMBER ENCODINGS

Figure 9 shows the structure of number embeddings learned on different datasets for different encodings. For P10 the models learn rotary structure which is reminiscent of other works such as grokking (Power et al., 2022), and allows recovering relative numbers from inner products. It is also interesting to see how different datasets can lead to different learned encoding structures, for instance the arithmetic tasks seem to induce a more precise curve structure, while the planet data leads to more spread out embeddings, perhaps because the task is less sensitive to small perturbations of the numbers.

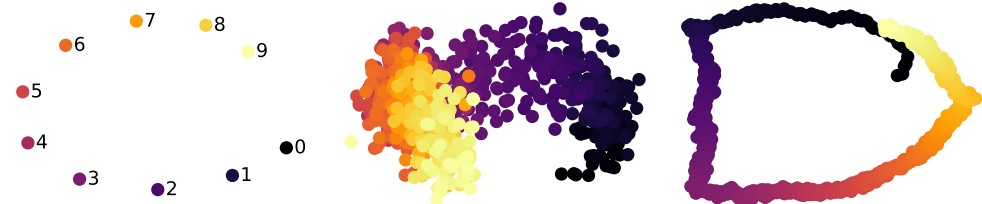

Figure 9: Two-dimensional PCA projection of the learned embeddings for mantissa tokens. (left) P10 encoding trained on the planet dataset; (center) P1000 encoding trained on the planet dataset; (right) P1000 encoding trained on the arithmetic dataset. Brighter colors denote higher number values.

### B.6 A NUMBER LOOK-UP EXPERIMENT

In order to explore potential deleterious effects of using XVAL which encode the values of the numbers multiplicatively we set up a simple number lookup experiment.[4] In this problem, we train the model on textual samples that demonstrate a dictionary lookup task:

```
"{d:1.53, e:-1.33, a:2.53, i:0.0232} e=-1.33"
```

In this experiment, we sample the number between -3 and 3 but we withhold a small region between 0.3 and 0.5 such that they do not appear in the dataset. As with our other experiments, we train a transformer model using self-supervised learning, in this case mask-filling with a masking probability of 30%.

In this experiment, we evaluate two metrics. The first is the Mean Square Error between the looked up number and the actual number. The second is the accuracy of reconstructing the number exactly as it appeared in the dictionary. Because our comparison textual number encodings have only 3 significant figures, we evaluate the accuracy of this lookup up to 3 significant figures and also provide the 2 significant figure accuracy for comparison. The results of the in-distribution and out of distribution evaluations are given in Tabs. 11 and 12. XVAL performs well on MSE but is unable to reconstruct the number exactly. The text encodings do better on accuracy but because of outliers, their performance on the MSE suffers. Furthermore, the text based encodings' performance drastically degrades on the held-out set whereas XVAL's does not degrade to the same extent.

**Note.** Because of the residual connections of the transformer, the information regarding the number value is principle present at readout time. Because of this, the network can in principle recover the value in context.

Table 11: Performance of the different encodings on the number lookup experiment. FP15 provides the best accuracy and XVAL provides the best MSE.

| Method | MSE | Accuracy | |
| --- | --- | --- | --- |
| | | 3 Sig Figs | 2 Sig Figs |
| P10 | $(2.1 \pm 0.4) \times 10^{-2}$ | $91\% \pm 3\%$ | $94\% \pm 2\%$ |
| FP15 | $(7 \pm 0.7) \times 10^{-3}$ | $99.9\% \pm 0.1$ | $99.9\% \pm 0.1\%$ |
| XVAL | $(3 \pm 0.5) \times 10^{-3}$ | $6\% \pm 1\%$ | $55\% \pm 5\%$ |

---

[4]We are grateful to reviewer iRZg for suggesting looking into cases where a language model needs to reference a previously mentioned number in context.

Table 12: Out of distribution performance of the different encodings on the number lookup experiment. Neither textual encoding schemes generalize to unseen values.

| Method | MSE | Accuracy | |
| --- | --- | --- | --- |
| | | 3 Sig Figs | 2 Sig Figs |
| P10 | $(3.8 \pm 0.3) \times 10^{-1}$ | $65.7\% \pm 2\%$ | $66.5\% \pm 1\%$ |
| FP15 | $(0.9 \pm 0.4) \times 10^{-1}$ | $34.5\% \pm 0.5$ | $34.5\% \pm 0.5\%$ |
| xVAL | $(3.5 \pm 0.5) \times 10^{-3}$ | $1.5\% \pm 1\%$ | $12\% \pm 2\%$ |