# OpenReview forum: "xVal: A Continuous Number Encoding for Large Language Models"
_ICLR.cc/2024/Conference — Submitted to ICLR 2024_

### Official Review · Reviewer_aAmt · 2023-10-29

**Soundness:** 1 poor
**Presentation:** 1 poor
**Contribution:** 1 poor
**Rating:** 1
**Confidence:** 4

**Summary:**

The paper proposed a two-step prediction for floating numerics embedded in the NLG tasks for the LM.

**Strengths:**

I seldom write this but for this paper it's hard for me to find one.

**Weaknesses:**

1. Not well motivated. The paper says it's LMs have historically struggled to solve simple arithmetic problems  but somehow many Chain-of-Thoughts paper contradicts the claim. There is no discussion and not literature review enough for this part.

2. The method itself is not very interesting.

3. Evaluation is rough, what are FP15, P10 and so on? No clear elaboration on this.

4. Page 2 is in low-resolution. A not ready draft IMO.

**Questions:**

I believe at least the paper need to show empirically why the task it difficult.

---

> ### Author Response · Authors · 2023-11-21
> **Response to reviewer aAmt**
>
> We thank the reviewer for their feedback. Here are our responses to the points raised by the reviewer. We kindly ask the reviewer to reconsider their evaluation in light of the results and the improved performance that our encoding offers.
>
> **Weaknesses**
>
> > 1. Not well motivated. The paper says it's LMs have historically struggled to solve simple arithmetic problems but somehow many Chain-of-Thoughts paper contradicts the claim. There is no discussion and not literature review enough for this part.
>
> Indeed, a number of recent work have explored methodologies that force LLMs to use Chain-of-Thought reasoning in a “scratchpad” setting and have proven effective for improving numerical predictions of LLMs tasks like arithmetic or mathematical word problems. We have now added this information and relevant citations to the introduction.
>
> However, a pre-requisite for these methods to be applicable is that Chain-of-Thought reasoning for solving the given task (or a similar task) must be available during train time. In our work, we are motivated by the need to improve numerical analysis of datasets such as those seen in many scientific contexts. For these tasks, such chain-of-thought reasoning is not known -- indeed often the goal is to discover the relationships in the first place -- and therefore a step-by-step scratchpad would not improve the results.
>
> []()
>
> > 2. The method itself is not very interesting.
>
> We believe that our findings and results would be of interest to the community as evidenced by the comments from other reviewers. In particular, we show state-of-the-art out-of-distribution generalization performance in a number of tasks.
>
> []()
>
> > 3. Evaluation is rough, what are FP15, P10 and so on? No clear elaboration on this.
>
> These tokenization schemes (FP15, P10 etc) are described in detail in Table 1 (including an example for clairty) and at the beginning of Section 3 titled **Comparison with other number encodings**.
>
> []()
>
> > 4. Page 2 is in low-resolution. A not ready draft IMO.
>
> Thank you for pointing out this accidental glitch in the rendering process. We have corrected this in our latest version.
>
> []()
>
> **Questions**
>
> > I believe at least the paper need to show empirically why the task it difficult.
>
> In our experiments, we compare xVal with existing text-based number encoding schemes that are currently SOTA on the type of numerical analysis tasks that we are interested in. We demonstrate that these other schemes have several notable weaknesses, especially in terms of robustness to outliers and performance in out-of-distribution generalization.

---

### Official Review · Reviewer_qrLU · 2023-10-31

**Soundness:** 3 good
**Presentation:** 3 good
**Contribution:** 3 good
**Rating:** 6
**Confidence:** 3

**Summary:**

Large language models rely on predefined tokenizers to process input data. While commonly used tokenizers are constructed for natural language, it is challenging to apply them to numbers. In response, the authors propose a novel method to encode number values for large language models.

Specifically, at model input, the author proposes to incorporate the numerical value of numbers as a weighted sum of token embedding and position embedding; at model output, the author proposes to construct a separate number head to decide the numerical value and use the original token head to decide whether to use this token.

The author conducts extensive experiments, which demonstrates the effectiveness of the proposed method.

**Strengths:**

1. The studied problem is important and may have a big impact. The proposed method is reasonable and novel.
2. The author conducts empirical evaluations on: a) learning arithmetic; b) temperature forecasting; c) planetary orbit prediction. The proposed method demonstrates consistent performance gain.

**Weaknesses:**

In experiment setting, the proposed method is only evaluated in the supervised training setting. It is unclear on the impact of the proposed method on pre-training tasks.

**Questions:**

How the training hyper-parameters are configured and why different encodings have different configurations.

---

> ### Author Response · Authors · 2023-11-21
> **Response to reviewer qrLU**
>
> We thank the reviewer for their feedback. We are encouraged that they find the problem important and our method potentially impactful. Below we address the weakness and question brought up by the reviewer. We have added new experiments and clarified parts of the text that we believe have overall improved the manuscript.
>
> **Weaknesses**
>
> > In experiment setting, the proposed method is only evaluated in the supervised training setting. It is unclear on the impact of the proposed method on pre-training tasks.
>
> Thank you for bringing this up. While our training regimen is self-supervised masked language modeling, we primarily evaluate the performance of the model on values that are directly present in the training corpus. We suspect that this is what the reviewer is referring to as "evaluated in the supervised training setting".
>
> To more directly evaluate the model in a setting where the output is not explicitly present in the input, we performed two new tests. The first a fine-tuning task and the second a roll-out task.
>
> * We fine-tuned the temperature forecasting model on a binary downstream task, specifically we use the fine-tune the pretrained model to predict if the first reporting station is located on water or on a landmass. In principle, this problem is solvable by looking at the latitude & longitude of the station, but because we fine-tune on only 500 samples, there is not enough information to accurately learn the mapping between latitude & longitude and water coverage over the earth. Therefore, the model needs to use other information (e.g. temperature variation etc.) to make this inference. We find that xVal outperforms the other models, achieving a ROC AUC of 0.620 ( vs .600 and 0.580 for P10 and FP15 respectively). We expect that the longer context length of P10 is adversely affecting its fine-tuning behavior. We report these findings in Sec. B.1.3.
>
> * In the temperature forecast problem, we rolled-out the temperature forecast on 5 extra time-steps to predict the future development of the temperature across all sites. This is similar to text generation in language models that are only trained to predict the next time-step. We find again that xVal outperforms the other encoding schemes. This is not surprising since xVal has the best one-step prediction performance.
>
> **Questions**
>
> > How the training hyper-parameters are configured and why different encodings have different configurations.
>
> Thank you for asking this clarifying question. To determine the hyper-parameters of our runs, we performed a coarse grid search for the learning rate (with a linear grid in log space). We chose the best performance on a validation set and reported the results on an unseen test set. In many cases, the validation performance of different learning rates were very close and/or within variance of each other, but we still chose the configuration with the best performance. This is one cause of differing learning rates between different experiments. Another reason for differences between the optimal hyperparameters for xVal and the text encodings is the very different statistics of the input for these different models. Whereas the input to the text models is always a text embedding, the input to xVal cares about the numbers that appear in the text.
>
> We have added this description to the supplementary materials Sec. B.

---

### Official Review · Reviewer_iRZg · 2023-10-31

**Soundness:** 3 good
**Presentation:** 3 good
**Contribution:** 2 fair
**Rating:** 5
**Confidence:** 3

**Summary:**

This paper introduces an innovative numerical encoding scheme, designed to efficiently represent any real number using a single token. The encoding leverages a dedicated embedding vector, denoted as `<NUM>`, which is dynamically scaled by the numerical value. This approach significantly optimizes token usage and minimizes the vocabulary footprint.

In addition, the authors complement this encoding scheme with a novel number-inference technique, incorporating a specialized `Number Head`. This `Number Head` enables the model to generate continuous real numbers in an end-to-end manner.

To validate the effectiveness of this proposed methodology, extensive evaluations were conducted on both synthetic and real-world datasets. The results demonstrated consistently comparable or superior performance when compared to prior research in the field.

**Strengths:**

The strengths of this paper are as follows:

1.  The paper introduces a deceptively simple yet novel approach to real number representation. This design not only minimizes token usage but also significantly reduces vocabulary footprint while preserving the input's value. This simplicity is an attractive feature, emphasizing efficiency without sacrificing performance.

2.  The proposed method exhibits outstanding performance, particularly in synthetic datasets used to evaluate multi-digit multiplication and multi-operand binary tree combining. The results indicate that it excels at preserving real number information, outperforming previous approaches in these scenarios.

3.  The paper is meticulously organized, presenting the idea in a coherent and transparent manner. It guides the reader through the experimental process, offering a step-by-step validation of the proposed method and effectively highlighting the distinctions from various baseline techniques.

4.  One notable advantage of this approach is its adaptability to out-of-distribution inputs. This robustness is inherent in the generation of embeddings, allowing the method to handle cases where certain real numbers are more frequently predicted due to their prevalence or distribution discrepancies between training and testing datasets.

**Weaknesses:**

This paper, despite its strengths, has some weaknesses:

1.  An issue with the rendering quality on page 2, affecting some figures, diminishes the readability of both the text and data points. The compromised legibility of axis labels and data points could potentially hinder the reader's comprehension and impact the overall impression of the paper.

2.  The paper's discussion of implicit normalization and its impact on real number embedding output is not sufficiently clear. The authors fail to provide a lucid explanation of how layer normalization influences the output of real number embeddings and why the normalization into a specific range is performed during preprocessing. Moreover, it remains unclear how these aspects might affect the performance of baseline methods. A more comprehensive and intuitive explanation is required to enhance the paper's accessibility.

3.  The experiments conducted in the paper exclusively utilize structured data in JSON format, focusing on scenarios like multi-digit multiplication and multi-operand calculations. While these experiments demonstrate the effectiveness of the proposed approach in these specific contexts, they do not adequately showcase the method's capability to understand real numbers in the broader context of natural language. This limitation may raise questions about the universal applicability and effectiveness of the proposed approach. Expanding the scope of experiments to encompass real-world language contexts would provide a more comprehensive evaluation of its capabilities.

**Questions:**

1.  Could you please demonstrate how well the proposed method can handle situations where it needs to refer to previously mentioned real numbers in the context, ensuring these numbers remain unaltered? How does this embedding method impact a Language Model's capability to preserve real numbers in the given input?

2.  How is the capability of the proposed numerical encoding scheme affected by extremely small or extremely large numbers? Is it able to maintain representation accuracy and robustness in the presence of such numerical extremes?

3.  Can you provide an example of a use case where the proposed method demonstrates improved real number understanding capabilities, but the input data is not structured as in JSON, a binary tree, or multi-digit multiplication? This would help illustrate the method's applicability in contexts beyond structured data scenarios.

---

> ### Author Response · Authors · 2023-11-21
> **Response to reviewer iRZg**
>
> We thank the reviewer for their insightful feedback. We are encouraged that they find our approach simple and well explained. We are also pleased that the reviewer finds the improved generalization of our method noteworthy.
>
> Thanks to the reviewer's suggestions, we have added a new experiment and also further clarifying comments to the manuscript. We kindly ask the reviewer to reconsider their evaluation in light of these additions.
>
> In this post, we address the weaknesses and in the next we respond to the questions.
>
> []()
>
> ## Weaknesses
> []()
>
> > 1. An issue with the rendering quality on page 2, affecting some figures, diminishes the readability of both the text and data points. The compromised legibility of axis labels and data points could potentially hinder the reader's comprehension and impact the overall impression of the paper.
>
> Thank you for pointing out this accidental glitch in the rendering process. We have corrected this in our latest version.
>
> []()
>
> > 2. The paper's discussion of implicit normalization and its impact on real number embedding output is not sufficiently clear. The authors fail to provide a lucid explanation of how layer normalization influences the output of real number embeddings and why the normalization into a specific range is performed during preprocessing. Moreover, it remains unclear how these aspects might affect the performance of baseline methods. A more comprehensive and intuitive explanation is required to enhance the paper's accessibility.
>
> **Impact of Layer-Norm:** We have added clarifying text in Sec. 2.1 to provide more information about the effect of the layer normalization. In particular,
>
> “In our experiments, we use additive positional encodings and therefore the result of the layer-norm is to normalize the sum of the vector associated with the [NUM] token and the positional encoding vector.”
>
> However, as we point out in our ablation tests in Table 9 of the supplementary materials, the presence of the first layer norm does not affect the results of our experiments. (Removing all layer norms, on the other hand, causes the performance to degrade.) We believe this is because of the pre-processing rescaling, which makes the first layer-norm unnecessary.
>
> **Impact of Rescaling during Pre-Processing:** As we mention in the draft, the limited dynamic range of xVal is a drawback of the current implementation of our algorithm. Because of this, the range of the input numbers had to be rescaled such that it is compatible with this limited dynamic range. We have verified that this preprocessing rescaling step does not affect the performance of the textual baseline methods. But this step was necessary for the non-text based comparisons in the Appendix (e.g. MLPs of Sec. B.1.2). Also note that (as mentioned in the conclusion) the limited dynamic range of our method can be improved by using more complex embedding schemes that maintain the continuity of the embedding scheme. However, we decided to use the simplest continuous  encoding scheme so as not to distract the reader by the details of a higher dynamic range encoding.
>
> []()
> > 3. The experiments conducted in the paper exclusively utilize structured data in JSON format, focusing on scenarios like multi-digit multiplication and multi-operand calculations. While these experiments demonstrate the effectiveness of the proposed approach in these specific contexts, they do not adequately showcase the method's capability to understand real numbers in the broader context of natural language. This limitation may raise questions about the universal applicability and effectiveness of the proposed approach. Expanding the scope of experiments to encompass real-world language contexts would provide a more comprehensive evaluation of its capabilities.
>
> We agree that it would be instructive to demonstrate the use of our encoding scheme in more general settings. The goal in this current work was to demonstrate better performance when applying LLMs to the analysis of numerically dense scientific data from heterogeneous sources.
>
> **On structure:** We note that even though our samples were formatted as a JSON, this formatting is not necessary for achieving the results in the draft. Violating this strict structure, for example removing structural components of the JSON (e.g. commas and brackets) or providing information out of place (e.g. putting a key in a different part of the JSON hierarchy) does not degrade the performance, so long as these were also seen during training. (In comparison, methods that rely on the strict structure of JSON would fail when the structure is violated.)
>
> **Broadening the scope:** Demonstrating better number understanding in real-world language contexts would be the logical next step. However, training a language model with broad language understanding requires dramatically more computation time and power than the experiments run in the current paper.

---

> > ### Author Response · Authors · 2023-11-21
> > **Response to questions**
> >
> > In this post we address the questions raised by the reviewer.
> >
> > ## Questions
> >
> > > 1. Could you please demonstrate how well the proposed method can handle situations where it needs to refer to previously mentioned real numbers in the context, ensuring these numbers remain unaltered? How does this embedding method impact a Language Model's capability to preserve real numbers in the given input?
> >
> > We thank the reviewer for this suggestion. We address this as follows.
> >
> > **New Experiment:** In response, we have carried a new experiment demonstrating how a language model trained with xVal can indeed recall numbers. The experiment and explanations can now be found in Sec. B.6 of the Supplementary Materials, and we summarize them below:
> >
> > We implemented a simple number lookup experiment where each sample is a combination of a dictionary and an example of a dictionary lookup. An example is “{d:1.53, e:-1.33, a:2.53, i:0.0232} e=-1.33”. We train this via self-supervised masked language modeling and then evaluate the performance of the lookup task on unseen samples. We find that with xVal the model can perform the number lookup with very good MSE on the target number, but often fails to recover the exact number to 3 digit precision. (We also train this task using FP15 and P10 and provide the comparison in Sec. B.6).
> >
> > To see whether or not this performance generalizes to number ranges unseen in the training set, we evaluate out of distribution performance of the number lookup task. We again find that xVal generalizes better compared to P10 and  FP15.
> >
> >
> > **Architectural Perspective:** From an architectural perspective, the model can access the numbers present in the context because of the residual connections which connect the input embedding directly to the final output of the transformer blocks. Therefore, in principle the transformer can make sure that this information remains available by leaving that subspace of the latent space unchanged.
> >
> > []()
> >
> > > 2. How is the capability of the proposed numerical encoding scheme affected by extremely small or extremely large numbers? Is it able to maintain representation accuracy and robustness in the presence of such numerical extremes?
> >
> > We discuss this in Section 2.1 (“xVal: A Continuous Number Encoding”). As shown in Figure 2, the input dynamic range of xVal is indeed limited, particularly outside of the range [-5, 5]. Before training, we therefore preprocess our data to scale the values to fall within this range.
> >
> > We discuss the shortcoming of the limited dynamic range of our proposed method in the Conclusion section. We expect that that this shortcoming can be alleviated through more elaborate continuous encodings.  (Also see our response to Weakness #2 above.) However, we believe that by focusing on the simplest continuous encoding (i.e. our multiplicative implementation) we can highlight the core advantages - as well as disadvantages - of this approach more clearly.
> >
> > []()
> >
> > > 3. Can you provide an example of a use case where the proposed method demonstrates improved real number understanding capabilities, but the input data is not structured as in JSON, a binary tree, or multi-digit multiplication? This would help illustrate the method's applicability in contexts beyond structured data scenarios.
> >
> > As we wrote in our response to Weakness #3 above, removing or corrupting the strict JSON formatting structure does not degrade the overall performance of xVal, though in some experiments we saw that the structure makes it easier for the model to converge more quickly. We have verified this empirically, for example by replacing the full JSON structure by simple CSV formatting. Testing the model’s performance in the context of natural language, however, is outside of the scope of this current result, as it would require considerably more resources.

---

> > > ### Comment · Reviewer_iRZg · 2023-11-22
> > > **Thanks for the response**
> > >
> > > Thanks for the responses, I would raise my score to 5 given my concern on layernorm is settled yet the other concerns are still there.

---

### Official Review · Reviewer_SWrp · 2023-11-02

**Soundness:** 3 good
**Presentation:** 3 good
**Contribution:** 2 fair
**Rating:** 6
**Confidence:** 3

**Summary:**

This paper introduces a simple approach to encoding numerical values as tokenized input to a LLM. Specifically, all numbers $x$ in the input are identified and replaced with the stand-in token '[NUM]', that is then scaled by the value of $x$, i.e., $h(x) := x h(\text{[NUM]})$. This both reduces the number of tokens per number and the vocabulary size, and leads to more efficient training. The paper also demonstrates improved performance at numerical tasks.

**Strengths:**

The method is simple and easy to understand. It also has some computational benefits. While on simple arithmetic tasks the model performs similarly well to other good approaches (e.g., on 3-5 digit multiplication, P1000 and B1999 encoding schemes can also get near perfect performance), xVal seems to work much better on unstructured, numerical heavy experiments. There are some shortcomings (some of which the authors make a good point of noting), but in general, it seems like a straightforward and effective representation strategy for numerically-dense text.

**Weaknesses:**

While the continuous nature of the xVal embedding can obviously be an advantage in some domains, I'm not sure how well it would work in general. For example, tasks like summarization, or question answering, where numerical values such as years/dates/account numbers are not meant to be worked with in the sense of arithmetic or other mathematical operations but simply carried about may lose performance. That said, for domain specific applications (like science), this may not be an issue. A hybrid approach may also work (e.g., representing $7.4$ as "+ , 740, e-2, 7.4 * [NUM]").

**Questions:**

- I'm not entirely sure why the runtime is so dramatically reduced. Is this due to the reduced length of each input/target and the vocab size? If the latter is a big factor, I'm surprised that what seems like a fairly small additional overhead of the softmax size for everything but FP15 setting would make that big of a difference.

- I'm curious as to what kind of empirical range the xVal style network has. Is $h(\texttt{[NUM]})$ layer normalized at the end of the network? Is the mapping from $h(\texttt{[NUM]})$ to its numerical value by way of MSE loss minimization linear? I'd imagine that the output range would be restricted in this setup.

- I'm a bit put off by the dependence on _parsing_ numerical quantities accurately, especially in messier settings. Curious if that posed any difficulties.

---

> ### Author Response · Authors · 2023-11-21
> **Response to reviewer SWrp**
>
> We thank the reviewer for their encouraging remarks and insightful questions. In this post, we respond to the weaknesses and in the next we address the questions mentioned.
>
> **Weaknesses**
>
> > While the continuous nature of the xVal embedding can obviously be an advantage in some domains, I'm not sure how well it would work in general. For example, tasks like summarization, or question answering, where numerical values such as years/dates/account numbers are not meant to be worked with in the sense of arithmetic or other mathematical operations but simply carried about may lose performance. That said, for domain specific applications (like science), this may not be an issue. A hybrid approach may also work (e.g., representing  as "+ , 740, e-2, 7.4 * [NUM]").
>
> We agree that for a general language model a hybrid approach would be useful. As the reviewer suggests, an approach combining text-based encoding with a continuous encoding to leverage advantages of each method can lead to improvements on both. One can also envision an encoding scheme that treats different number types (integers, floats, etc) differently.
>
> In this work, we wanted to demonstrate the advantages of a continuous number encoding over text-based encodings in the setting of heterogeneous scientific data analysis. We leave the exploration of combining different tokenization schemes to future work.

---

> ### Author Response · Authors · 2023-11-21
> **Response to questions**
>
> **Questions**
>
> > * I'm not entirely sure why the runtime is so dramatically reduced. Is this due to the reduced length of each input/target and the vocab size? If the latter is a big factor, I'm surprised that what seems like a fairly small additional overhead of the softmax size for everything but FP15 setting would make that big of a difference.
>
> The reduced runtime of xVal is a combination of both fewer tokens and small vocabulary size. The bigger factor is the length of the tokenized input. Text-based encodings (other than FP15) have dramatically increased number of tokens compared to xVal and FP15  (in Sec. 3 around 8000 and 5000 tokens per sample respectively for P1000 and P10 compared to 1800 tokens for FP15 and xVal (see Table 1 for a summary of different tokenization lengths). This leads to a longer runtime for the encodings with higher number of tokens per number. In comparison, the overhead of FP15 compared to xVal is coming from the size of the vocabulary leading to large embedding and logit-head layers. Note that the relative increase in computational cost from the embedding and logit-head is larger for small transformer models. In our case, the embedding matrix of FP15, for example, is a large fraction of the total number of parameters of the model.
>
> Note that in the runs with equal number of total tokens, for efficient parallelization purposes, the number of tokens seen per batch is still different for the different methods. This also leads to a difference in run-time, even when the total number of tokens seen during training is the same.
>
> []()
>
> > * I'm curious as to what kind of empirical range the xVal style network has. Is $h\texttt{[Num]}$ layer normalized at the end of the network?
>
> We use an architecture similar to GPT such that the layer-norms of the network are only placed in transformer blocks (prior to the attention module and the MLP) and the residual path through the blocks does not go through layer-norm. There are no other layer-norms in the network.
>
> []()
>
> > * Is the mapping from to its numerical value by way of MSE loss minimization linear? I'd imagine that the output range would be restricted in this setup.
>
> In the majority of our experiments we use a non-linear map (MLP with one hidden layer)  to extract the numbers from the final output of the transformer blocks before evaluating the MSE loss. However, in ablation tests we find that a linear readout layer performed equally well up to run-to-run variations.
>
> If, as proposed in the conclusion, we replace the current multiplicative encoding with a different continuous encoding with higher dynamic range, we also solve the problem of the output range. For example, we can use Fourier encodings on the log space of the input numbers, leading to a vector encoding with each component taking values between -1 and 1. The dynamic range of such an encoding can be easily extended and MSE loss minimization on this embedding would also be easily implemented. We chose to implement the simplest continuous encoding in the current draft in order to highlight the advantages of continuity and leave questions regarding extended dynamic range to future work.
>
> []()
>
> > * I'm a bit put off by the dependence on parsing numerical quantities accurately, especially in messier settings. Curious if that posed any difficulties.
>
> In our code, the preprocessing step is implemented using a short regular expression to parse the input text and replace numerical values with the “[NUM]” token. This is done efficiently using the Re package and the processing adds little overhead. This simple preprocessing step would not be sufficient for a general purpose language model, especially if we want to extract the number types as suggested above. However, for such a use case, lightweight parts of speech tagging models can be used to efficiently and accurately tag numbers and number types (e.g. Jurafsky Speech and Language Processing Chapter 8).

---

> ### Comment · Reviewer_SWrp · 2023-11-22
>
> Thanks for taking the time to respond to my review and questions.
>
> - I still have the same reservation about this being a good general purpose number representation (e.g., for copying numbers by value and by representing large magnitude numbers), as also shared by reviewer iRZg. Still, I agree that there are domains where this is less important, and, it would be interesting to see if a hybrid approach could bridge that gap.
>
> - I also recognize that numbers can be parsed using regular expressions and/or taggers, but these still have errors in messy situations. Again, however, I feel that it is still an OK contribution if this number representation is meant to be more domain specific.
>
> - One question about the layer norm discussion: I am assuming you are representing the vector $xu + p$ after layer normalization as
>
> $$ \mathrm{LayerNorm}(xu + p) = \frac{xu + p - \mathbb{E}[xu + p]}{\sqrt{\mathrm{Var}[xu + p]}}= \frac{xu + p'}{||xu + p'||}$$
>
> where $p' = p - \mathbb{E}[xu + p]$? Then $u \cdot p' \neq 0$. Some clarity here on your approach and analysis would be good.

---

> > ### Author Response · Authors · 2023-11-22
> >
> > > * I still have the same reservation about this being a good general purpose number representation (e.g., for copying numbers by value and by representing large magnitude numbers), as also shared by reviewer iRZg. Still, I agree that there are domains where this is less important, and, it would be interesting to see if a hybrid approach could bridge that gap.
> > > * I also recognize that numbers can be parsed using regular expressions and/or taggers, but these still have errors in messy situations. Again, however, I feel that it is still an OK contribution if this number representation is meant to be more domain specific.
> >
> > We agree on both counts. However, we believe that there is growing interest in the use of language models as versatile data analysis tools in heterogenous data settings. In these cases, the use-case is specialized (as the reviewer points out) and preprocessing and tagging expressions are straightforward.
> >
> > []()
> >
> > > * One question about the layer norm discussion: I am assuming you are representing the vector $xu + p$ after layer normalization as  $ \mathrm{LayerNorm(xu + p)} = \frac{xu + p - \mathbb{E}[xu + p]}{\sqrt{\mathbb{V}[xu + p]}} =  \frac{xu + p'}{||xu + p'||}$ where $p' = p - \mathbb{E}[xu + p]$? Then $u \cdot p' \neq 0$. Some clarity here on your approach and analysis would be good.
> >
> > This is indeed the formulation of layer-norm we are using which is default torch layer-norm behavior without a bias. We can slightly simplify this:
> >
> > 1. First, since the average is carried out per token, we have $\mathbb{E}[xu + p] =  x \mathbb{E}[u] + \mathbb{E}[p]$. This is because $x$ is an overall per-token scaling. We have
> > $$ \mathrm{LayerNorm(xu + p)} = \frac{x (u-\mathbb{E}[u]) + p - \mathbb{E}[p]}{\sqrt{\mathbb{V}[xu + p]}}$$
> >
> > 2. Second, we can simplify the expression using the fact that  $\mathbb{E}[u]$ and $\mathbb{E}[p]$ are small (compared to the components of these vectors). This is true at initialization because $u$ and $p$ are high dimensional Gaussian random variables but we have also verified it to be true empirically after training. For simplicity in this response we assume $\mathbb{E}[u]=\mathbb{E}[p]=0$. We have
> > $$ \mathrm{LayerNorm(xu + p)} \approx \frac{x u + p}{\sqrt{x^2 |u|^2 + |p|^2 + 2 u\cdot p}}$$
> >
> > 3. Next, if we assume that $u\cdot p$ is small (which is the case in our trained models), up to first order we have
> > $$ \mathrm{LayerNorm(xu + p)} \approx \frac{x u + p}{\sqrt{x^2 |u|^2 + |p|^2}} \qquad (*)$$
> > If for simplicity, we assume $|u|=|p|=1$, then the projection of the output of the layer-norm along the [NUM] direction becomes:
> > $$ \frac{u}{|u|}\cdot \mathrm{LayerNorm(xu + p)} \approx \frac{x}{\sqrt{x^2 + 1}}$$
> > as reported in the paper.
> >
> > **Note 1.** We made simplifying assumptions here that 1. $\mathbb{E}[u]$ and $\mathbb{E}[p]$ are small and 2. $u\cdot p$ is small. These assumptions are theoretically valid at initialization because of the high-dimensional nature of these random vectors but we have empirically verified that these hold after training, and that the final relation $(*)$ holds to better than $\pm 2$\%.
> >
> > **Note 2.** We show in the ablation tests in supplementary materials (Sec. B.4) that removing this initial layer-norm in fact does *not* change the performance of our model. (In comparison if we remove all layer-norms, the performance of our models significantly degrade.) The inclusion of the explanation of layer-norm in the manuscript was not to highlight it as an important step, but it was an attempt to answer the question: "Is layer-norm compatible with the multiplicative number encoding?" (See Uri's public comment on the same subject.) We will adjust the manuscript to make this point - as well as the details of the above arguments -  clear.
> >
> > []()
> >
> > ---
> >
> > []()
> > We thank the reviewer for asking questions that helped improve the manuscript. We kindly ask the reviewer to consider raising their score if they find our answers clarifying and our results - especially regarding generalization and analysis of heterogenous numeric data - of interest to the community.

---

### Public Comment · ~Uri_Alon1 · 2023-11-10
**Question**

Thank you for the interesting paper!

Section 2.1 says that:
> In our implementation, the multiplicative embedding of XVAL is followed by the addition of a positional encoding vector and then a layer-norm in the first transformer block.

Then, the paper argues that applying the LayerNorm does not change the direction of the original `[NUM]` embedding.

I have a few of questions about the interaction with the LayerNorm layer:

1. As shown in [Brody et al., 2023](https://arxiv.org/pdf/2305.02582.pdf) (but easy to derive) - the norm of a vector after going through LayerNorm is always $\sqrt{d}$ (modulo the affine transforms learnable scalars $\gamma$ and $\beta$, [for example in this definition](https://pytorch.org/docs/stable/generated/torch.nn.LayerNorm.html)).

Thus, LayerNorm is invariant to the norm of a vector.
So, this implies (and simple to derive) that $\textrm{LayerNorm}\left(v\right) =\textrm{ LayerNorm}\left(x \cdot v\right)$ where $x \in \mathbb{R}$ is the scalar to be encoded, and $v \in \mathbb{R}^{d}$ is the embedding of `[NUM]`.

So, since the approach relies on the norm of the vector which is lost in LayerNorm, I'm surprised that this approach helps and does not completely forget the value of the input number.

2. As a follow up question, how is Figure 2 possible? After LayerNorm, I would expect all vectors to have the same norm.
Otherwise, if I'm missing something, what makes the values to saturate outside the range of $[-5,5]$?

3. Does that mean that xVal cannot represent numbers beyond the range $[-5,5]$?

Thanks and good luck!
Uri

---

> ### Author Response · Authors · 2023-11-10
> **Thanks for the Question!**
>
> Dear Uri,
>
> Thanks very much for your interest in our paper!
> We answer your questions between the lines below:
>
> >Section 2.1 says that:
> >> In our implementation, the multiplicative embedding of XVAL is followed by the addition of a positional encoding vector and then a layer-norm in the first transformer block.
>
> > Then, the paper argues that applying the LayerNorm does not change the direction of the original [NUM] embedding.
>
> Indeed, LayerNorm does not change the direction of the vector it receives as input. However, it receives the combination of scaled number token and positional embedding  x * [NUM] + [POS]
>
> > I have a few of questions about the interaction with the LayerNorm layer:
>
> > As shown in Brody et al., 2023 (but easy to derive) - the norm of a vector after going through LayerNorm is always $\sqrt{d}$ (modulo the affine transforms learnable scalars $\gamma$ and $\beta$, for example in this definition).
>
> > Thus, LayerNorm is invariant to the norm of a vector.
>
> This is correct.
>
> > So, this implies (and simple to derive) that $\textrm{LayerNorm}\left(v\right) =\textrm{ LayerNorm}\left(x \cdot v\right)$ where $x \in \mathbb{R}$ is the scalar to be encoded, and $v \in \mathbb{R}^{d}$ is the embedding of [NUM].
>
> The formula is correct, but we input $\textrm{LayerNorm}\left(x\mathbf v + \mathbf p\right)$ which creates a slight offset and leads to ultimately preserving the norm information.
>
> If, for simplicity, we assume the positional embedding to be of unit norm and orthogonal to [NUM] (but this is generalizable to any positional embedding not collinear with [NUM]), then the projection of the LayerNorm output onto [NUM] leads to a projection value of $x/(\sqrt{1 + x^2})$, thus preserving the value of $x$.
>
> See also the formula at the bottom of section 2.1.
>
> Intuitively: We are recovering the value of $x$ because it is put into relation with the size of the positional embedding. The larger the input $x$, the more then output of LayerNorm will be close to [NUM] , the lower it is, the closer the output will be to the positional embedding.
>
> > So, since the approach relies on the norm of the vector which is lost in LayerNorm, I'm surprised that this approach helps and does not completely forget the value of the input number.
>
> > As a follow up question, how is Figure 2 possible? After LayerNorm, I would expect all vectors to have the same norm.
>
> > Otherwise, if I'm missing something, what makes the values to saturate outside the range of $[-5,5]$?
>
> > Does that mean that xVal cannot represent numbers beyond the range $[-5,5]$?
>
> Effectively the function resulting from applying LayerNorm to the scaled number token offset by the positional embedding has the form $\frac{x + a}{\sqrt{(x + b)^2 + c^2}}$, where $a, b$ and $c$ are parameters computable from the positional encoding and the number vector, this function is always a compressive nonlinearity. Its shape is similar to, but less saturating than, tanh. Depending on the scaling $c$, saturation will occur sooner or later.
>
> We are working on multi-scale approaches to number embedding where scales can hand off responsibility to each other once their saturation limit becomes relevant.
>
> > Thanks and good luck!
> > Uri
>
>
> We hope this answers your questions and thanks again for your interest!

---

### Meta-Review · Area_Chair_7sa1 · 2023-12-08

**Metareview:**

First, as a logistical note, I want to say up-front that I am meta-reviewing this paper without taking reviewer aAmt's review into account because it is excessively short, excessively negative, and does not have sufficient content to be taken into consideration.

This paper addresses the issue that current tokenizers treat numbers like characters rather than ordinal values so that the model must learn the relationship between numbers from scratch. To address this, the paper proposes learning a single token for every number that means "number" and using a given number's value to elementwise multiple the embedding for the "number" token. This implies that the model must have a separate output head for scaling the embedding when the "number" token is generated. Ultimately this encoding scheme works quite well in tasks where 1) understanding the values of numbers is important, 2) the numbers can fall in a predetermined range, and 3) the numbers are reasonably spread out across the range. The second and third considerations stem from technical limitations with the method that are well-outlined from the paper, for example the existence of layer norm after the embedding layer. A final limitation of this approach, discussed in the reviews, is that it's not appropriate when numbers *aren't* ordinal, which is entirely possible. On the whole, I can see this method being used in very constrained settings where the above-mentioned restrictions hold. I don't think this would be a good replacement for number encoding in general. The reviewers mentioned in various places that they are working on alternative methods that address these limitations. Since the current method is relatively simple and the limitations prevent it from being appropriate in most settings, I'd suggest developing these more advanced encoding schemes and resubmitting this paper once more convincing and broad results are generated.

**Justification For Why Not Higher Score:**

The settings where this method would be appropriate are too constrained, and the paper would be stronger once the limitations have been addressed.

**Justification For Why Not Lower Score:**

N/A

---

### Decision · Program_Chairs · 2024-01-16

Reject